# Factors Associated with Discrepancy of Child-Adolescent/Parent Reported Quality of Life in the Era of COVID-19

**DOI:** 10.3390/ijerph192114359

**Published:** 2022-11-02

**Authors:** Elodie Jeanbert, Cédric Baumann, Anja Todorović, Cyril Tarquinio, Hélène Rousseau, Stéphanie Bourion-Bédès

**Affiliations:** 1UR4360 APEMAC, Health Adjustment, Measurement and Assessment, Interdisciplinary Approaches, School of Public Health, Faculty of Medicine, University of Lorraine, 54000 Nancy, France; 2Methodology, Data Management and Statistics Unit, University Hospital of Nancy, 54000 Nancy, France; 3Versailles Hospital, University Department of Child and Adolescent Psychiatry, 78157 Versailles-Le-Chesnay, France

**Keywords:** quality of life, KIDSCREEN-27, child self-report, parent proxy report, discrepancy, well-being, mental health

## Abstract

Billions of children/adolescents experienced unprecedented changes in their daily lives that impacted their health-related quality of life (HRQoL) during the first wave of the coronavirus disease 2019. The purpose of this study was to describe child-parent discrepancies in reporting on HRQoL and explore factors associated with such discrepancies at the end of the first lockdown in France. A cross-sectional study was conducted among French school-aged children from 8 to 18 years and their parents living in the Grand Est region in France during the first wave of the epidemic. The impact of individual, self-reported health status and environmental data on discrepant parent–child reports of HRQoL was assessed by multinomial multivariable logistic regression models. A total of 471 parent–child pairs were included. Among 50% of the discordant pairs, parents underestimated HRQoL more frequently than they overestimated it. Home location, social support score, children’s education level, parents’ education level, tensions and conflicts with neighbors reported by children, whether they had access to a garden, and parents’ professional activity were significantly associated with parental overestimation (_adjusted_OR from 2.08 to 11.61; *p* < 0.05). Factors associated with parental underestimation were children’s education level, SF12 score, home location, the child’s gender, parent’s level of education, the presence of noise in the residence reported by children, whether a household member was infected with COVID-19, whether they had access to a garden, and family structure (_adjusted_OR from 1.60 to 4.0; *p* < 0.05). This study revealed differences between child-reported and parent-reported HRQoL. The COVID-19 pandemic accentuated the discrepancies in observable dimensions and attenuated them in unobservable dimensions of HRQoL but did not impact the directional discrepancy; parents underestimated their child’s HRQoL more. These discrepancies appear to be explained by parent and child sociodemographic factors.

## 1. Introduction

Childhood is a specific developmental period and there has been an increased interest in children’s health-related quality of life (HRQoL) over the last decade [1]. HRQoL is a multidimensional construct with many sub-dimensions of subjective experience, including physical health, psychological state, social interaction, and school performance [2,3]. Therefore, many instruments have been developed to assess the child’s perception of their HRQoL. A systematic review of generic and disease-specific instruments identified almost a hundred instruments designed for children and adolescents by the year 2008 [4]. At least 30 generic instruments are available for measuring HRQoL. The best known are the Child Health Questionnaire [5], the Child Health and Illness Profile [6], the KIDSCREEN [7], the Pediatric Quality of Life Inventory [8], and the Youth Quality of Life [9]. Measurement of the HRQoL of children under the age of eight may be more difficult to obtain, and therefore proxy versions of the instruments can be used in younger children [10]. Nevertheless, children and parents may have different perceptions of the child’s HRQoL [11], because HRQoL is a subjective concept which may be prone to measurement bias when estimated by another person [12,13], and because self-reports and proxy-reports are not interchangeable [14]. Thus it is recommended to let an individual report on their own HRQoL, perhaps with the addition of a proxy version of the questionnaire [10]. Indeed, some instruments, both self and proxy-parent assessment, allow an individual to assess the level of HRQoL of another person, when the latter is not able to complete a self-questionnaire, using a proxy version [15,16]. Among the questionnaires that offer both a child and a parent version, the KIDSCREEN is the most widely used in the literature due to its good psychometric properties and its multilingualism, and it has been tested on 22 827 children with a response rate of 69% [17]. It is used in many different contexts such as child cancer survivors [18], children with chronic illnesses [19], children with hearing loss [20], but also in the general population [21,22].

Discrepancy between child-HRQoL reports and parent-proxy reports has been investigated in the literature. However, although KIDSCREN-27 has been extensively validated, few studies have examined the discrepancies in scores between parents and children in the general population. A systematic review of child-parent agreement on HRQoL reported greater agreement for observable functioning (e.g., physical HRQoL), and lesser for non-observable functioning (e.g., emotional or social HRQoL) [23]. It was also reported that agreement was better between parents and chronically sick children compared with healthy children and their parents [23]. A study using qualitative methods to explain these discrepancies showed that HRQoL score discrepancies between children and parents could be explained by children’s reasoning and response styles that tended to answer with extreme scores or based their response to a question on a single example [15]. Moreover, parents, especially mothers, underestimated their child’s HRQoL score [16,24].

Overall, factors found most frequently in the literature are gender and age of the children/adolescents. However, these discrepancies are not always in the same direction. For example, a general population study showed that parent–child agreement decreased with age [12]. Conversely, a study of children with congenital glaucoma showed that there were more discrepancies in younger children [13]. In the same way, studies of the HRQoL of children with newly diagnosed chronic diseases or who have received a cochlear implant showed that parents underestimated their child’s HRQoL and that these discrepancies decreased with the child’s increasing age [25,26]. These same studies also showed discrepancies in different directions depending on the gender of the children/adolescents. For example, the study in children with congenital glaucoma showed that there were more discrepancies for girls [13]. Conversely, the general population study showed that agreement was higher for girls [12]. However, other factors, apart from the gender and the age of the child, have also been found in the literature to be associated with discrepancies. For example, a study of children with cerebral palsy showed that discrepancies were also influenced by both child and parent characteristics including the presence of child pain, associated with fewer discrepancies, and child behavior problems and parental stress which were associated with more discrepancies [27]. A study in small children showed that weaker parent–child relationships and better social support of children were associated with more discrepancies [28]. Comorbidities have also been found to be factors associated with discrepancies in visually impaired children [29]. Moreover, the level and type of education of the children and the level of education of the parents were also found to be significantly associated with discrepancies, as was the number of hospital admissions for children with chronic diseases [30]. Another Swedish general population study of parent/child pairs using the KIDSCREEN-27 concluded that it was important to look at parental characteristics when studying discrepancy because they might be the source of discrepancies, for example the parent’s HRQoL. Indeed, there were fewer discrepancies between children/adolescents and parents for parents with a higher HRQoL score [31]. Moreover, a study using the KIDSCREEN-27 carried out in the general population in seven European countries including France has shown that the discrepancies differed across countries and that it was important to look for specific factors [32]. It would therefore be interesting to look for factors associated with discrepancies in the French general population and see if some factors might be associated with discrepancies related to the health context of COVID-19.

Indeed, during the period of lockdown related to the COVID-19 pandemic, the HRQoL of children and adolescents has been negatively impacted [33]. Children and adolescents faced massive changes in their daily lives, including home lockdown, school closures and social restrictions that can be incongruent with their developmental tasks [34]. The French Grand-Est region was the first French area affected by COVID-19 during the first wave and one of the two most affected regions in France.

Thus, this study aimed to:(a)Examine child-parent discrepancies in reporting HRQoL in a sample of French school-aged children from 8 to 18 years.(b)Investigate the potential factors associated with such discrepancies in the KIDSCREEN-27 scores.

## 2. Methods

### 2.1. Procedure

A cross-sectional analysis was conducted of data from the observational Feelings and Psychological Impact of the COVID-19 Epidemic on Children, Adolescents and their Parents in the Grand Est Area during the first wave (PIMS2-CoV19) study. A random selection of schools stratified on the level (primary school, middle school, high school), the school sector (general, technical, professional or mixed) and the status of the school (public or private) was carried out from an exhaustive database of schools of the Academy of Nancy-Metz (Grand Est region, France) communicated by the rectorate. Parent and child/adolescent pairs were recruited electronically by their email address, and they completed an anonymous online survey from 26 May to 6 July 2020, at the end of the first lockdown. The web-based survey took approximately 20 min to complete for parents and 10 min for children/adolescents. All participants provided online informed consent to participate in the study. The survey was anonymous to ensure the confidentiality and reliability of data. Approval for the study protocol was obtained from the Commissioner for Data Protection (Comité National Informatique et Liberté-registration 2220408).

### 2.2. Participants

All children and adolescents enrolled in grades 3 through 12 in a school in the Nancy-Metz academy (Grand Est region, France), and their parents were eligible to participate in the study. A data set of 471 child-parent pairs was obtained with 471 parents from 341 distinct households participating. Socio-demographic data are shown in Table 1. The children’s mean age was 12.9 years (SD = 2.9) with 53.5% female. The parents consisted of 290 females (85.0%) and they were mostly 40 to 49 years of age (63.6%). Most parents worked as employees (41.6%), managers and higher intellectual professionals (24.6%) or had intermediate professions (15.2%), and 11.7% of parents did not work. Over half of the parents had completed a minimum of two years of higher education (64.5%) and they were mostly working full time (63.6%).

### 2.3. Data Collection

For children and adolescents, sociodemographic data, data on living and learning conditions, and self-perceived HRQoL data were collected. For parents, sociodemographic data and living conditions were also collected and they were also questioned about their self-perceived HRQoL [35], their child’s perceived HRQoL and their levels of resilience [36], stress [37,38], anxiety [39], and social support [40,41]. The data collected and the questionnaires used are presented in Appendix A.

### 2.4. Statistical Analysis

#### 2.4.1. Judgment Criterion

The judgment criterion was the discrepancy between the self-rated and parent-rated HRQoL (health-related quality of life) scores of children/adolescents in each dimension of the KIDSCREEN-27 [7,42,43]. To measure the judgement criterion, an intermediate variable ∆ (parent-estimated child/adolescent HRQoL score—child-estimated HRQoL score) was created. Because of the lack of Minimal Clinically Important Difference (MCID) for the KIDSCREEN-27, we applied the recommendations which consisted in using the discrimination threshold of one-half standard deviation of the distribution of the variable ∆ [44]. The variable ∆ was a 3-modality variable defined as follows:

If ∆ < −1/2 σ then the discrepancy was in the sense of an underestimation of children’s HRQoL by their parents (noted UE: underestimation);

If −1/2 σ ≤ ∆ ≤ 1/2 σ then it was a match between the self-rated and parent-rated HRQoL scores of children/adolescents (reference modality: match);

If ∆ > 1/2 σ then the discrepancy was in the overestimation of children’s HRQoL by their parents (noted OE: overestimation).

The variable ∆ was calculated for each parent–child/adolescent pair in each dimension of the KIDSCREEN-27.

#### 2.4.2. Descriptive and Comparative Analyses

Continuous variables were described by the means and standard deviations; medians were used to dichotomize variables when applicable. Categorical variables were described by percentages. Variables were described in the full sample of children/adolescents and according to the 3 school levels (primary school, middle school and high school) and compared using both Fisher’s exact (categorical variables) and Mann–Whitney tests (continuous variables).

#### 2.4.3. Main Analyses

Multinomial logistic regression models were used to determine which variables were associated with discrepancies between children’s self-reported and parent-reported HRQoL scores in each dimension of the KIDSCREEN-27. Multinomial logistic regression is an extension of logistic regression to categorical variables with three or more modalities. In our study, the variable to be explained had 3 modalities and the reference modality was the agreement between both children/adolescent- and parent-rated HRQoL scores. Firstly, bivariable multinomial logistic regression models were used and variables with a *p*-value < 0.1 were candidates to multivariable multinomial logistic regression models. Pearson correlation, Phi coefficients and variation inflation factors (VIFs) were calculated to verify the lack of correlation and multicollinearity. Crude Odds Ratios (CORs), Adjusted Odds Ratio (AORs) and 95% confidence intervals (CIs) were estimated. For each model, the goodness-of-fit was assessed by calculating the model determination coefficient (R^2^) and the percentage that was predicted correctly by the model. Analyses were performed using SAS 9.4 (SAS Inst., Cary, NC, USA).

## 3. Results

### 3.1. Characteristics of Children/Teenager’s Sample

The sociodemographic, learning, and working characteristics of the 471 children/adolescents are presented in Table 2 for the entire sample and by school level groups. Of the 471 children/adolescents, 35.7% found it difficult or very difficult to focus at home for their homework and 27.8% found it difficult or very difficult to be home-schooled. Most children and adolescents reported spending more than 2 h per day on their homework (74.2%) and 80.1% reported not being afraid (or a little afraid) to go out and contract COVID-19. Almost a quarter of the sample (22.2%) reported never leaving home. Of the 471 children/adolescents, 2.3% felt there were tensions and conflicts with neighbors and 29.1% felt there were tensions and conflicts at home. Some children and adolescents had difficulty with isolating themselves at home (13.2%). The results did reveal significant differences according to the different types of school levels. Indeed, girls were more represented in the high school students than in the other school levels (*p* < 0.0001). Students at higher school levels had less difficulty with being home-schooled (*p* = 0.0267) and to focus at home for homework (0.0131). There were also statistically significant differences according to the different types of school levels regarding time spent doing homework (*p* = 0.0131) and fear of going out and contracting COVID-19 (*p* < 0.0001).

### 3.2. Characteristics of the Households and Parents’ Scores on the Various Questionnaires

Characteristics of the households and parents’ scores on the various questionnaires are presented in Table 3. Of the 341 households, most families lived in a house (80.0%), 61.4% lived in a rural area and 6.5% reported having no access to outdoor areas. Both parents lived in the same home for two-thirds (67.8%) of the households. Almost 10% of the parents were living in the same place of residence with someone who was a suspected case of COVID-19 and 3.0% with someone infected with COVID-19. The mean MSPSS total score was 5.5 (SD = 1.2). The mean GAD-7 score was 5.2 (SD = 4.7) with 18.5% of them experiencing moderate to severe anxiety. The mean SF-12 scores were 70.3 (SD = 12.7) and 56.6 (SD = 16.7) for the PCS and MCS domains, respectively. The mean PSS-10 score was 14.0 (SD = 7.8) with moderate stress for 44.3% of parents and high stress for 5.6% of parents, while 65 parents (19.1%) reported a low level of resilience.

### 3.3. Child and Parents’ Reports of HRQoL and Child–Parent Discrepancy

The mean HRQoL scores of the children and the parents’ ratings are shown in Table 4 for the entire sample and by levels of education. For the “autonomy and parents” and “peers and social support” dimensions, scores increased with the school level, whereas for the “physical well-being”, “psychological well-being” and “school environment” dimensions, scores decreased with the school level. The distribution of parent–child discrepancies on each dimension of the KIDSCREEN-27 is depicted in Figure 1. The children’s lowest average score was 36.4 ± 14.7 for the “peers and social support” dimension, while the highest average score was 48.8 ± 10.0 for the “psychological well-being” dimension. Significant differences were found between the full sample and the parents’ sample for three dimensions. All scores provided by the parents were lower than those provided by the children. The “physical well-being” dimension was the most affected with a discrepancy of 57.1%. Conversely, the “peers and social support” dimension was the least affected with a discrepancy of 48.2%. The most frequently overestimated of the children’s scores by parents was in the “school environment” dimension (24.2%) and the most frequently underestimated was the “physical well-being” dimension (41.4%).

### 3.4. Factors Associated with Child–Parent Discrepancies

Table 5 shows the results of the bivariable and multivariable multinomial logistic regression in each dimension of the KIDSCREEN-27. Only variables significantly associated with any HRQoL scores are presented in Table 4. We present the results of multivariable analyses only. FIVs were consistently <2, indicating a lack of multicollinearity. We will give example interpretations of the multivariable analysis for underestimation and overestimation. For the “psychological well-being” dimension, children’s gender was associated with underestimations (OR = 2.00 95% CI: 1.32–3.04). It can be interpreted as follows: compared to concordant scores, a boy was twice as likely to have his HRQoL score underestimated by his parents than a girl. Conversely, for the “peers and social support” dimension, home location was associated with overestimations (OR = 2.08 95% CI: 1.15–3.75). This can be interpreted as follows: using the “matching parent/child scores” modality as the reference, in families who lived in an urban area, parents were 2.08 times more likely to overestimate their child’s/adolescent’s HRQoL level compared to parents who lived in a rural area.

For the “physical well-being” dimension, children who reported having tensions and conflicts with their neighbors were associated with a higher likelihood of overevaluation by their parents compared to those who did not (OR = 5.04 95% CI:1.14–22.27). For parents, having a low mental health score was a risk factor for underestimating their children’s HRQoL (OR = 1.71 95% CI: 1.13–2.58). Parents who lived in a rural area were more likely to underestimate their child’s HRQoL, than parents who lived in an urban area (OR = 1.76 95% CI: 1.15–2.69). Having someone suspected of being affected by COVID-19 at home was a risk factor for parents to underestimate the HRQoL of their children (OR = 2.69 95% CI: 1.32–5.47).

For the “psychological well-being” dimension, the male gender of children was associated with a higher probability of parental under evaluation (OR = 2.00, 95% CI: 1.32–3.04). Compared to parents who were searching for job, studying, retired or disabled, parents who worked full-time and parents who worked half or part-time had a higher probability of overestimating their child’s HRQoL score (OR = 6.28, 95% CI: 1.43–27.53, OR = 11.61 95% CI: 2.52–53.63, respectively). In terms of social support for parents under lockdown, a high “significant other” score was associated with over-assessments by parents compared to a score above the median (OR = 2.38 95% CI: 1.40–4.04).

For the “autonomy and parents” dimension, children who reported that there was noise in the residence were associated with a higher likelihood of under evaluation by their parents compared to those who did not (OR = 2.45 95% CI: 1.17–5.12). Compared to parents who had no diploma, or who had a primary school certificate, secondary school diploma, CAP or BEP or equivalent, parents who had a baccalaureate, vocational certificate or equivalent were associated with a higher probability of overvaluation (OR = 3.26 95% CI: 1.26–8.41) and under evaluation (OR = 2.05 95% CI: 1.00–4.25). Finally, children who lived with both their parents or children who lived with a single parent were more likely to have their HRQoL scores underestimated by their parents than children who lived with one of their parents and his or her partner (OR = 2.50 95% CI = 1.19–5.27 and OR =4.00 95% CI = 1.70–9.40, respectively).

For the “peers and social support” dimension, children who were in middle or high school were more likely to have their HRQoL scores overestimated (OR = 3.85 95% CI: 1.85–8.03, OR = 3.24 95% CI: 1.45–7.26, respectively) and underestimated for middle school (OR = 1.60 95% CI: 1.01–2.53) by their parents than children who were in primary school. Parents who lived in an urban area were more likely to overestimate their child’s HRQoL than parents who lived in a rural area (OR = 2.08 95% CI: 1.15–3.75).

For the “school environment” dimension, compared to children who had access to a private outside space, children who had no access to a private outside space were more likely to have their HRQoL score underestimated by their parents (OR = 3.09 95% CI = 1.20–7.95) and children who had a courtyard or garden for collective use were more likely to have their HRQoL score overestimated by their parents (OR = 5.89 95% CI = 1.54–22.45). For parents, having a low “physical health” score was a risk factor for underestimating their children’s HRQoL (OR = 1.77 95% CI: 1.14–2.75).

## 4. Discussion

Our study showed that about 50% of parent–child score pairs were concordant regardless of the dimension. Parents tended to underestimate their children’s HRQoL (health-related quality of life) level. These results appear to be consistent with previous studies that reported that, in general, parents tended to underestimate the HRQoL of their children [12,13,32]. In our study, the discrepancy between child-reported and parent-reported HRQoL persisted in each school level with parents giving lower scores, on average, than children in all dimensions of the KIDSCREEN-27. The COVID-19 pandemic had no additional impact on the direction of the discrepancies. Indeed, there were already more underestimations of the HRQoL of children by parents than underestimations in studies conducted outside the pandemic period [12,13,32]. The size of the child-parent gap increased with the school level in “autonomy and parents” and “peers and social support” dimensions and decreased in others. This might be because adolescence is a time of rapid biological, social, and emotional change that can affect child-parent relationships and social support [27]. In addition, the quality of peer relationships and social support perceived by children may often not be visible to the parents, especially in adolescence [11] and in an anxious period such as the lockdown related to the COVID-19 pandemic [45]. In contrast, the domains of physical well-being, psychological well-being, and school environment might reflect more factual and objective functioning in adolescence, resulting in a smaller gap [27].

Discrepancies between children and parents were higher when children’s HRQoL scores were low and lower when HRQoL scores were high. For example, in the “peers and social support” dimension, the mean children’s HRQoL score was the lowest and there were 3.5 points of difference with the score estimated by parents, representing the largest average gap between parents and children. In this dimension, the discrepancies between parents and children were the highest, but it was also in this dimension that there were the most matching pairs. The reason why the parent–child gap was higher in this dimension may be because the study was conducted during a period of lockdown related to the COVID-19 pandemic. Indeed, mean child/adolescent HRQoL scores were lower in all dimensions than in the reference validation study, which was not conducted during the lockdown period. In all dimensions, the scores varied but remained generally good compared to the reference study in four of the five dimensions [7]. However, for the “peers and social support” dimension, the scores varied greatly and were alarming during the period of lockdown, compared to the reference study. Indeed, in the lockdown period, scores were 36.4 ± 14.7 versus 49.9 ± 10.02 in the reference study (*p* < 0.0001). This finding was consistent with another study that points out that loneliness resulting from lockdown was particularly problematic for children and adolescents [46]. The children’s HRQoL scores in the “peers and social support” dimension during the lockdown being very low, compared to an unconfined period, could explain the large gap with the parents’ scores. Moreover, there was a score effect; the lower the level of the children’s HRQoL, the greater the gap between parents and children, but the worse the children’s scores were, the less discrepancies there were and therefore parents noticed it more.

Our results showed that the lowest agreement rates were present in the dimensions of “physical well-being” and “school environment”. Both dimensions could be considered as observable dimensions of HRQoL for parents that were not related to the child’s internal experiences. These results were contrasted with previous research, which described better agreement for observable dimensions [11,12,47]. The COVID-19 pandemic may have had an impact on the degree of discrepancy by accentuating the discrepancies in the observable dimensions compared to studies conducted outside of the COVID-19 period in the general population [11,12] or in sick children [47].

Similarly, in this period of the pandemic, these observable dimensions were also more affected by undervaluation and overvaluation than other dimensions. Indeed, it was in the “physical well-being” dimension that parents underestimated the HRQoL of their children the most. Under-reporting of children’s HRQoL scores by parents in this dimension were more frequent when someone in the home was suspected of having COVID-19. It is possible that when someone in the home was suspected of having COVID-19, parents thought their children were physically unwell when in fact the children were feeling physically better. This is why the COVID-19 pandemic may have accentuated undervaluation and discrepancies in general in an observable dimension such as physical well-being. In the same way, the COVID-19 pandemic may have had an impact that increased discrepancies in the “school environment” observable dimension because during this period schooling was entirely done at home and parent may have had difficulty “judging” the HRQoL of children related to this dimension because it was a discovery for all during this period. It was represented by the highest percentage of overestimates and also a high percentage of underestimates.

Conversely, the COVID-19 pandemic attenuated discrepancies in unobservable dimensions such as the “psychological well-being” dimension where discrepancies were more numerous in the non-pandemic period [11,12,47]. Concerning the “psychological well-being” dimension, the older the children, the lesser the gap between child and parent. This can be explained by the fact that younger children might have more difficulty expressing their emotional needs to their parents than older children [13]. In this dimension, the child’s gender was associated with discrepancies between both children and parent-rated HRQoL scores of the children; being a girl was a protective factor for undervaluation. Girls might be more vocal about their health and feelings than boys, they might be more concerned about the coronavirus which may manifest more frequently because they might have less ability to cope with it. This may explain the attenuation of the discrepancy in this unobservable dimension.

For the “peers and social support” dimension, the HRQoL scores increased with the school level. In fact, adolescents had a higher HRQoL score in the “peers and social support” dimension than children. There were 10.9 points of difference between HRQoL scores of primary school students and the HRQoL scores of high school students. It can come from the fact that the teenagers, in general, have phones as early as in middle school, and are more and more connected, especially with regards to social networks, and can thus keep contact with their friends or family, even during a period of lockdown related to the COVID-19 pandemic.

This study has some limitations, including the limitations of the PIMS2-CoV19 study. First, mothers were over-represented in the study (85.0%) which limited the representativeness even though this is not uncommon in pediatric health care studies and clinical practice [48]. Moreover, 80% of families lived in a house, which also limited the representativeness. Secondly, the parent–child pairs were from a single region, and this was one of the French areas that was the most substantially affected by COVID-19. This limited the generalization of the results. However, it would be reasonable to consider it as a minor bias because all of France was confined and schools were closed in all regions, so all children stayed at home under the lockdown, even if their region wasn’t affected by COVID-19 as much. Moreover, the generalization of the results is complicated for another reason. Indeed, because of the absence of MCID for KIDSCREEN-27, the threshold chosen for the discrepancies depended on the characteristics of the sample (one-half standard deviation from the mean of ∆), so the generalization could be difficult for a different sample.

Among the strengths of the study, this was, indeed, the first study to examine discrepancies between children’s self-rated HRQoL scores in the general population in France and to the best of our knowledge, this was the first study to have been conducted among children/adolescents and their parents during the first COVID-19 pandemic wave and thus allows us to see the impact of the pandemic on the degree of parent–child agreement. The second strength is that our statistical analysis was based on a clear definition of the discrepancy and the use of a generic instrument to assess the HRQoL with child and parents’ versions available. Moreover, as the level of concordance of the HRQoL of an individual assessed by another person is not always optimal, this study provides recommendations for the use of parent’s data in the study of child HRQoL and for better interpretation of the parent-estimated child HRQoL scores in the future studies. At the same time, there is a need to take precautions with the interpretation of the discrepancies since the study was carried out during the lockdown related to the COVID-19 pandemic, which could perhaps lead us to a “bounded effect”, impacting the results and limiting or accentuating the discrepancies. It will be necessary to proceed to a possible weighting of the scores in future studies in relation to the factors found to be associated with discrepancies. In view of the results of our studies, and in particular the 50% discrepancy found in each of the dimensions, it will be necessary to integrate measures of discrepancy in the validation of questionnaires in self and parent-proxy evaluation. This will allow us to raise awareness of the social isolation of the youngest, and to integrate preventive measures in case of a similar situation.

## 5. Conclusions

To conclude, there were differences between child-reported and parent-reported quality of life. The COVID-19 pandemic accentuated the discrepancies in observable dimensions of HRQoL and attenuated them in non-observable dimensions of HRQoL. In contrast, the directional discrepancy has not been impacted by the COVID-19 pandemic, and showed parents’ tendency to underestimate their child’s HRQoL. Moreover, factors associated with the discrepancies were also not impacted by the pandemic and were parent- and child-related sociodemographic factors or factors related to living conditions. It is therefore important to take these factors into account when using parent-estimated child HRQoL in future studies.

## Figures and Tables

**Figure 1 ijerph-19-14359-f001:**
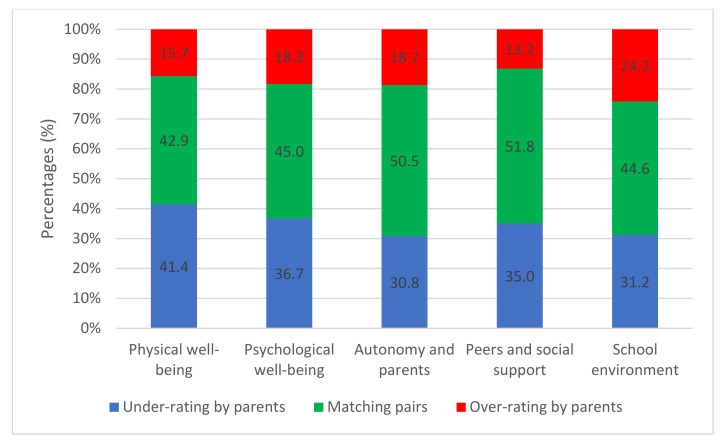
Distribution of matching and mismatching pairs in the different dimensions of the KIDSCREEN-27.

**Table 1 ijerph-19-14359-t001:** Socio-demographic characteristics of parents and children.

	Children N = 471 N (%)/Average (SD)	Parents N = 341 N (%)
**Gender**		
Male	219 (46.5)	51 (15.0)
Female	252 (53.5)	290 (85.0)
**Age**		
8–11	188 (39.9)	
12–14	121 (25.7)	
15–18	162 (34.4)	
<30		2 (0.6)
30–39		85 (25.0)
40–49		217(63.6)
>50		37 (10.8)
**Socio-professional category**		
Farmers-operators		1 (0.3)
Artisans, shopkeepers and company managers		8 (2.3)
Managers and higher intellectual professions		84 (24.6)
Intermediate professions		52 (15.2)
Employees		142 (41.6)
Workers		10 (2.9)
Retired		4 (1.2)
Other persons not working		40 (11.7)
**Highest diploma**		
No diploma/primary school certificate/secondary school certificate, CAP, BEP or equivalent		56 (16.4)
Baccalaureate, vocational certificate or equivalent		65 (19.1)
Higher education diploma (minimum 2 years of higher education)		220 (64.5)
**Professional activity**		
Full-time		214 (63.6)
Half-time/part-time		80 (23.7)
Job search/studies		29 (8.6)
Disability/retirement		15 (4.5)

Abbreviation: SD, Standard Deviation.

**Table 2 ijerph-19-14359-t002:** Sociodemographic data, learning and working conditions of the children/adolescent’s sample.

	Full Sample N = 471 N (%)/Average (SD)	Primary School Students N = 187 N (%)/Average (SD)	Middle School Students N = 171 N (%)/Average (SD)	High School Students N = 113 N (%)/Average (SD)	*p-*Value
**Gender** Male Female Age	219 (46.5) 252 (53.5) 12.9 (2.9)	92 (49.2) 95 (50.8) 9.9 (0.8)	94 (55.0) 77 (45.0) 13.5 (1.24)	33 (29.2) 80 (70.8) 17.0 (0.87)	**<0.0001**
**Focus at home for homework** Not at all difficult A little difficult Difficult Very difficult	143 (30.5) 159 (33.9) 94 (20.1) 73 (15.6)	53 (28.3) 51 (27.3) 44 (23.5) 39 (20.9)	55 (32.4) 64 (37.6) 28 (16.5) 23 (13.5)	35 (31.3) 44 (39.3) 22 (19.6) 11 (9.8)	**0.0431**
**Homework time** +4h/day every day Between 2h and 4h/day every day -2h/day every day -1h/day every day	137 (29.4) 209 (44.8)97 (20.8) 23 (4.9)	39 (21.0) 98 (52.7) 42 (22.6) 7 (3.8)	57 (33.7) 64 (37.9) 39 (23.1) 9 (5.3)	41 (36.9) 47 (42.3) 16 (14.4) 7 (6.3)	**0.0131**
**Have to be home-schooled** Not at all difficult A little difficult Difficult Very difficult	197 (42.2) 140 (30.0) 75 (16.0) 55 (11.8)	77 (41.6) 48 (25.9) 35 (18.9) 25 (13.5)	79 (46.5) 51 (30.0) 17 (10.0) 23 (13.5)	41 (36.6) 41 (36.6) 23 (20.5) 7 (6.3)	**0.0267**
**Fear of going out and having the COVID-19** Not at all A little A lot Enormously	210 (44.9) 165 (35.2) 55 (11.7) 38 (8.1)	79 (42.2) 48 (25.7) 34 (18.2) 26 (13.9)	82 (48.5) 67 (39.6) 14 (8.3) 6 (3.6)	49 (43.8) 50 (44.6) 7 (6.3) 6 (5.4)	**<0.0001**
**Exit outside the dwelling** Several times a day, almost every day Once a day, almost every day Several times a week but not every day Approximately once a week Less than once a week Never	51 (10.9) 91 (19.4) 77 (16.4) 59 (12.6) 87 (18.6) 104 (22.2)	24 (12.8) 34 (18.2) 32 (17.1) 24 (12.8) 30 (16.0) 43 (23.0)	21 (12.3) 36 (21.1) 27 (15.8) 19 (11.1) 31 (18.1) 37 (21.6)	6 (5.4) 21 (18.9) 18 (16.2) 16 (14.4) 26 (23.4) 24 (21.6)	0.6982
**Noises outside the residence** Yes No	54 (11.5) 417 (88.5)	24 (12.8) 163 (87.2)	16 (9.4) 155 (90.6)	14 (12.4) 99 (87.6)	0.5518
**Noises inside the residence** Yes No	43 (9.1) 428 (90.9)	17 (9.1) 170 (90.9)	16 (9.4) 155 (90.6)	10 (8.8) 103 (91.2)	0.9892
**Tensions or conflicts with neighbors** Yes No	11 (2.3) 460 (97.7)	4 (2.1) 183 (97.9)	4 (2.3) 167 (97.7)	3 (2.7) 110 (97.3)	0.9597
**Tensions and conflicts at home** Yes No	137 (29.1) 334 (70.9)	47 (25.1) 140 (74.9)	55 (32.2) 116 (67.8)	35 (31.0) 78 (69.0)	0.3017
**Difficulty isolating at home** Yes No	62 (13.2) 409 (86.8)	20 (10.7) 167 (89.3)	25 (14.6) 146 (85.4)	17 (15.0) 96 (85.0)	0.4532

Abbreviation: SD, Standard Deviation.

**Table 3 ijerph-19-14359-t003:** Household characteristics and parents’ scores on the various questionnaires.

	Parents	
	N = 341	
	N	%/Average (SD)
Parents’ scores		
**MSPSS total score**	341	5.5 (1.2)
**MSPSS subscales**		
Family	341	5.5 (1.3)
Friends	341	5.3 (1.4)
Significant other	341	5.7 (1.2)
**GAD-7 total score**	341	5.2 (4.7)
Normal anxiety (0–4)	170	49.9
Average anxiety (5–9)	108	31.7
Moderate anxiety (10–14)	47	13.8
High anxiety (15–21)	16	4.7
**PSS-10 total score**	341	14.0 (7.8)
Low stress (0–13)	171	50.1
Moderate stress (14–26)	151	44.3
Severe stress (27–40)	19	5.6
**BRS total score**	341	3.6 (0.8)
<3	65	19.1
≥3	276	80.9
**SF12 mental health score**	341	56.6 (16.5)
**SF12 physical health score**	341	70.3 (12.7)
Households		
**Type of accommodation**	340	
Apartment/mobile home	68	20
House	272	80
**Access to a private outside space**	341	
Private balcony, courtyard or terrace	38	11.1
Private domestic garden	270	79.2
Courtyard or garden for collective use	11	3.2
No access	22	6.5
**Someone at home had COVID-19**	341	
Confirmed and hospitalized cases	3	0.9
Confirmed and non-hospitalized cases	7	2.1
Suspected cases	32	9.4
No	299	87.7
**Home location**	339	
Urban area	131	38.6
Rural area	208	61.4
**Family structure**	339	
Single parent	62	18.3
Original couple	230	67.8
Parent + partner	47	13.9

Abbreviation: SD, Standard Deviation; MSPSS, Multidimensional Scale of Perceived Social Support; GAD-7, Generalized Anxiety Disorder scale 7; PSS-10, Perceived Stress Scale; SF12, item-12 Short Form survey.

**Table 4 ijerph-19-14359-t004:** Children’s quality of life scores assessed by themselves and by their parents.

	Full Sample N = 471 Average (SD)	Primary School Students N = 187 Average (SD)	Middle School Students N = 171 Average (SD)	High School Students N = 113 Average (SD)	Parents N = 471 Average (SD)	*p-*Value
**Dimension 1: Physical well-being**	45.9 (10.3)	49.4 (9.6)	44.9 (10.5)	41.8 (9.4)	42.9 (9.4)	**<0.0001**
**Dimension 2: Psychological well-being**	48.8 (10.0)	51.2 (9.2)	48.2 (10.3)	45.7 (9.8)	46.9 (11.3)	**0.0054**
**Dimension 3: Autonomy and parents**	47.7 (11.3)	46.3 (9.6)	47.8 (11.8)	49.7 (12.7)	46.2 (12.2)	0.0610
**Dimension 4: Peers and social support**	36.4 (14.7)	31.5 (15.8)	37.8 (12.6)	42.4 (13.3)	32.9 (14.9)	**0.0003**
**Dimension 5: School environment**	48.2 (10.2)	50.0 (9.8)	47.0 (10.1)	46.9 (10.4)	47.2 (10.5)	0.1275

Abbreviation: SD, Standard Deviation; *p*-value is the comparison between the full sample and the parents’ sample.

**Table 5 ijerph-19-14359-t005:** Factors associated with discrepancy between children’s quality of life scores assessed by themselves and their parents in each dimension of the KIDSCREEN-27 (N = 471).

	Bivariable Multinomial Logistic Regression			Multivariable Multinomial Logistic Regression		
	OR	95% CI	*p*-Value	OR	95% CI	*p*-Value
**Dimension 1: Physical well-being**				R^2^ = 0.08, H&L = 0.41		
Children: **Tensions and conflicts with neighbors** (Yes vs. No) Parents: **SF12 Mental health score** (Ref: > Median score) **Home location** (Rural area vs. Urban area) **Someone at home had COVID-19** (Ref: No) Suspected cases	UE: 1.04 OE: 4.81 UE: 1.58 OE: 1.14 UE: 1.68 OE: 1.35 UE: 2.53 OE: 1.11	0.21–5.20 1.12–20.65 1.06–2.34 0.67–1.95 1.11–2.53 0.78–2.35 1.27–5.03 0.38–3.24	0.9652 **0.0347** **0.0244** 0.6320 **0.0134** 0.2810 **0.0081** 0.8461	0.95 5.04 1.71 0.99 1.76 1.41 2.69 0.93	0.18–4.91 1.14–22.27 1.13–2.58 0.56–1.75 1.15–2.69 0.80–2.50 1.32–5.47 0.29–2.99	0.9515 **0.0329** **0.0110** 0.9722 **0.0088** 0.2364 **0.0063** 0.9050
**Dimension 2: Psychological well-being**				R^2^ = 0.1, H&L = 0.71		
Children: **Gender** (Male vs. Female) Parents: **Professional activity** (Ref: Job search /studies /disability /retirement) Full time Half-time/part-time **MSPSS significant others score** (Ref: ≤ Median score)	UE: 1.91 OE: 1.13 UE: 0.93 OE: 3.75 UE: 1.48 OE: 7.01 UE: 0.92 OE: 2.20	1.27–2.86 0.68–1.87 0.51–1.71 1.10–12.84 0.74–2.96 1.95–25.24 0.60–1.41 1.32–3.66	**0.0019** 0.6413 0.8254 **0.0349** 0.2648 0.0029 0.7040 **0.0024**	2.00 1.07 0.81 6.28 1.32 11.61 0.95 2.38	1.32–3.04 0.63–1.82 0.44–1.51 1.43–27.53 0.65–2.68 2.52–53.63 0.61–1.47 1.40–4.04	**0.0011** 0.7984 0.5163 **0.0148** 0.4441 **0.0017** 0.8049 **0.0014**
**Dimension 3: Autonomy and parents**				R^2^ = 0.08, H&L = 0.96		
Children: N**oises inside the residence** (Yes vs. No) Parents: **Highest degree** (Ref: No diploma/primary school certificate/secondary school diploma, CAP, BEP or equivalent) Baccalaureate, vocational certificate or equivalent **Family structure** (Ref: Parent + partner) Single parent Original couple	UE: 2.41 OE: 2.05 UE: 2.15 OE: 2.61 UE: 3.88 OE: 1.18 UE: 2.48 OE: 0.86	1.17–4.98 0.88–4.81 1.08–4.31 1.07–6.37 1.67–9.01 0.51–2.70 1.19–5.17 0.45–1.65	**0.0171** **0.0978** **0.0303** **0.0346** **0.0016** 0.7012 **0.0157** 0.6487	2.45 2.16 2.05 3.26 4.00 1.00 2.50 0.82	1.17–5.12 0.92–5.10 1.00–4.25 1.26–8.41 1.70–9.40 0.42–2.41 1.19–5.27 0.42–1.60	**0.0172** 0.0784 **0.0496** **0.0147** **0.0015** 0.9903 **0.0160** 0.5686
**Dimension 4: Peers and social support**				R^2^ = 0.06, H&L = 0.93		
Children: **Education level** (Ref: Primary school) Middle school High school Parents: **Home location** (Urban area vs. Rural area)	UE: 1.59 OE: 3.72 UE: 1.69 OE: 3.87 UE: 1.50 OE: 2.16	1.01–2.51 1.79–7.70 1.01–2.83 1.76–8.51 1.00–2.26 1.22–3.82	**0.0471** **0.0004** **0.0453** **0.0008** **0.0521** **0.0081**	1.60 3.85 1.55 3.24 1.45 2.08	1.01–2.53 1.85–8.03 0.92–2.62 1.45–7.26 0.95–2.21 1.15–3.75	**0.0466** **0.0003** 0.1001 **0.0042** 0.0834 **0.0153**
**Dimension 5: School environment**			R^2^ = 0.07, H&L = 0.71			
Parents: **SF12 Physical health score** (Ref: > Median score) **Access to a private outside space** (Ref: Private domestic garden Courtyard or garden for collective use No access	UE: 1.77 OE: 0.99 UE:<0.001 OE: 5.80 UE: 2.65 OE: 1.69	1.15–2.71 0.63–1.57 <0.001–>999.99 1.53–21.96 1.08–6.53 0.59–4.81	**0.0087** 0.9862 0.9812 **0.0096** **0.0334** 0.3240	1.77 1.11 <0.001 5.89 3.09 1.90	1.14–2.75 0.69–1.79 <0.001–>999.99 1.54–22.45 1.20–7.95 0.65–5.60	**0.0108** 0.6517 0.9811 **0.0095** **0.0192** 0.2414

Abbreviations: UE, Under-estimation of children’s quality of life scores by parents; OE, Over-estimation of children’s quality of life scores by parents, SF12, item-12 Short Form survey; MSPSS, Multidimensional Scale of Perceived Social Support; OR, odds ratio; Reference modality: match between child and parent.

## Data Availability

The data collected and analyzed during the current study are available from the corresponding author upon request.

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
