# Peer review of "Factors Associated with Discrepancy of Child-Adolescent/Parent Reported Quality of Life in the Era of COVID-19"

_ijerph, 2022, doi:10.3390/ijerph192114359_

Round 1

Reviewer 1 Report

This is a very interesting and relevant paper, reporting significant findings related to health-related quality of life (HRQoL) for school children in grades 3 to 12. A range of measures (KIDSCREEN-27) related to HRQoL were taken, from 471 child-parent pairs in a cross-sectional study, and correlation with possibly relevant factors were examined using multinomial logistic regression. The methods used were suitable and sound and are well described. The presentation of results in tables and figures was generally clear and suitable. Table 1 was a little unclear to me, as regards the gender and age of children. Details of gender for children seem to be missing, and the ages listed are only appropriate for adults, so the age distribution of the children cannot be seen? The meaning of the figures in the second column (Children) was not clear to me. Table 1 should be corrected.

The whole paper is well written, clear and logically presented. Shortcomings of the study are not overlooked, and the work is clearly related to the existing literature – which is quite comprehensively presented. The study was conducted in relation to the COVID pandemic, a time of known negative impact on HRQoL, which makes it of particular importance. The research is impressive because of the large number of participants and the breadth of measures and factors considered. A particular focus was on discrepancies between child and parent estimates of HRQoL, and the findings add to what is known about this key aspect of family life during difficult situations. The discussion was accurate to the results and well-reasoned.

Reviewer 2 Report

Dear authors,

thank you for your important work. The manuscript is well-written and easy to comprehend. In my opinion, the title promises a lot, but the results do not meet the expectation that the reader has from the title.

Introduction:

The underlying relevance of the topic was embedded in a very short theoretical background. A more detailed introduction into the several factors (other than gender/age) that may influence the discrepancy between parents and children would be desirable.

Line 46. Your statement about the reliability of assessment of HRQOL with children is very strict and internationally incorrect. There are several studies including reliable and valid measures of HRQOL with children from the age of 5 (e.g. Varni+, 2007: doi: 10.1186/1477-7525-5-I). This works very well with the five-year-olds, also from my own experience. Hence, I would not write this sentence so restrictively.

Line 49. Too much space.

Line 52. Term “hetero-assessment“ - You use the term “proxy or parent assessment” throughout the whole manuscript. I would stay consistent here. Beside, I my opinion this term is more unusual in the clinical/ HRQOL research field.

Line 64-65, 80-81. Be more precise in citation. Split the cited literature for each argument. Here: age vs gender.

Line 68, 74. Reference is missing.

Line 71. Which characteristics? It’s unclear.

Line 77. Doubled term – Health-related + HRQoL

Methods/Results

Line 127. Check English. “mean age was 12.9 years”. Without “old”.

Line 132. Table 1. The column of the table “Children” seems to be not correct, e.g., the gender of the children is missing. 219 children were < 30 years of age? Etc.

Line 186. Table 2. It is unclear what the p-values refer to by its shifted position in the table.

Line 192. Table 3. Some values in the right column are shifted, e.g., PSS-10, SF12,…

Line 204. Table 4. Some values are shifted.

Line 258. Table 5. Some values are shifted.

I don't like the fact that only tables are shown in the result section and the explanations can only be found in the appendix.

Discussion

Line 316. In my point of view, this is the essential fact in this study, but it was referred to the appendix again (but again, the argumentation in the appendix is not the expected one: what is the impact of the pandemic on the agreement?). At least that is what the title promises. As a reader, I would like to know how the pandemic affected the quality of life in contrast to the reference group. However, the data for the reference group are not listed at all in the statistical design and come unexpectedly in the discussion section. However, this belongs in the results section.

The entire discussion is a description of the results rather than a controversial, interesting integration of HRQoL in the pandemic. A substantiated argumentation would be desirable. Likewise, the data of the reference group should be included in the results section so that the reader can also read and classify these results in comparison to the lockdown-group. The impact of the pandemic is almost fully missing. What are the differences to other studies about HRQoL (without pandemic, but with children with specific diseases for example)? What can we conclude? With which arguments?

Round 2

Reviewer 2 Report

Dear authors,

well done. Thank you for improving the manuscript. The highlighting of the aim as well as the thematic embedding in existing literature is now well done, mainly in the introduction and discussion part.

Just minor things:

- line 386: double points

- line 399: start sentence with capitalization

Author Response

Dear Reviewer, 

Thank you very much for your kind comment. Indeed, there must have been a problem in the formatting. The problem with the double points has been solved (line 378 of the word and pdf versions that we resubmitted). Regarding the capitalization problem, the sentence started with "indeed" in our version and is located on line 342 of the resubmitted manuscript. All the problems have been solved. We hope that it will meet your expectations.